# Immune Response and Serum Biomarker Screening in Pregnant Women with Influenza A Virus Infection: Insight into Susceptibility and Disease Severity

**DOI:** 10.3390/diseases13060182

**Published:** 2025-06-10

**Authors:** Suping Zhang, Jiarun Jiang, Rui Wang, Yuanyuan Zhang, Jinghui Sun, Wanting Hong, Likai Qi, Jia Zang, Zeyi Liu, Yu Xia, Haibing Yang, Liling Chen

**Affiliations:** 1Suzhou Center for Disease Control and Prevention, Suzhou 215131, China; spzhang012@163.com (S.Z.); 302110120952@stu.xzhmu.edu.cn (R.W.); yuanyuan_zhang2025@163.com (Y.Z.); sunjinghui@stu.njmu.edu.cn (J.S.); wanting_hong@stu.njmu.edu.cn (W.H.); qilikai1989@163.com (L.Q.); jiazang@zju.edu.cn (J.Z.); xiayusuzhou@hotmail.com (Y.X.); yhbing111@163.com (H.Y.); 2School of Public Health, Xuzhou Medical University, Xuzhou 221004, China; 303110120949@stu.xzhmu.edu.cn; 3School of Public Health, Nanjing Medical University, Nanjing 211166, China; 4Institue of Respiratory Diseases, Soochow University, Suzhou 215006, China; zeyiliu@suda.edu.cn; 5Department of Respiratory Medicine, The First Affiliated Hospital of Soochow University, Suzhou 215006, China

**Keywords:** influenza A virus, pregnant women, immune dysfunction, TBNK cells, cytokines, Trp–Kyn pathway

## Abstract

Background: Seasonal influenza infection poses substantial risks to pregnant women, yet the immunological mechanisms underlying their heightened disease susceptibility remain incompletely characterized. Methods: This study employed multiparametric immunophenotyping and metabolic profiling to investigate cellular immunity, cytokine dynamics, and serum biomarkers in pregnant women infected with H3N2 across gestational stages. Through integrated flow cytometric analysis of peripheral blood mononuclear cells (PBMCs), multiple cytokine quantification, and LC-MS-based serum metabolomics, we compared immunological parameters, serum cytokines, and metabolites across trimesters in pregnant women infected and not infected with H3N2. Results: The results revealed reduced CD4^+^/CD8^+^ T cell ratios, a diminished CD27^+^ memory B cell population in pregnant women infected with H3N2, and elevated NK cells and Th2-skewed cytokines (IL-4, IL-6, IL-10) in severe influenza cases. Metabolomic profiling identified the dysregulation of the tryptophan–kynurenine (Trp–Kyn) pathway, with a 15-fold increase in the Kyn/Trp ratio in severe influenza compared to a normal pregnancy as a potential biomarker. Conclusions: These results elucidate synergistic pathophysiological axes-immune dysregulation and tryptophan metabolism alteration that potentially drive adverse outcomes. The identified biomarker panel (CD4/CD8 ratio, IL-6, Kyn/Trp ratio) shows potential clinical promise for early risk stratification in high-risk pregnancies with influenza infection.

## 1. Introduction

Seasonal influenza epidemics cause substantial global morbidity and mortality, disproportionately affecting pregnant women, and contributing to a significant economic burden [1,2]. Despite transient declines during COVID-19 (potentially driven by immune liabilities), influenza resurgence underscores the need for targeted prevention strategies [3,4]. At present, influenza A virus subtypes H1N1, H3N2, and influenza B virus Victoria lineage remain the primary drivers of seasonal influenza outbreaks [5].

Influenza virus infections can lead to acute respiratory illness with the potential for severe complications, particularly in high-risk populations such as pregnant women [6]. Research consistently demonstrates a significantly elevated hospitalization rate among pregnant women during influenza pandemics and seasonal outbreaks compared to non-pregnant women [7,8]. This vulnerability translates to increased incidence and mortality rates from influenza infection in pregnant and postpartum women, underscoring the heightened risk of severe illness in this population [9,10].

Physiological adaptations during pregnancy, particularly immune tolerance and dynamic hormonal fluctuation, may heighten vulnerability to influenza virus infection and disease severity [11]. However, the mechanistic relationship between pregnancy-induced immune remodeling and viral pathogenesis remains inadequately defined. Current diagnostic approaches for severe influenza predominantly rely on clinical symptom recognition [12], which often coincides with established cellular and molecular pathology. This diagnostic delay underscores an urgent need to establish validated early-warning biomarkers capable of identifying high-risk pregnant individuals during initial symptomatic phases.

This study aims to investigate the complex interactions between immune cell alterations, cytokine profiles, and serum biomarkers in pregnant women infected with the influenza virus. By elucidating the mechanisms of immune dysregulation and increased viral susceptibility in this population, our study will provide a scientific basis for developing targeted public health strategies to prevent and manage influenza infections in pregnant women in the post-pandemic era.

## 2. Materials and Methods

### 2.1. Patient Recruitment and Sample Collection

This study received ethical approval from the institutional review board of the Chinese Center for Disease Control and Prevention (approval number: 202014-01-02). We conducted a prospective cohort study from October 2023 to March 2024, enrolling pregnant women aged 20–40 years presenting with acute respiratory infections at sentinel hospitals in Suzhou City through the “Suzhou Area Maternal Influenza Related Acute Respiratory Infection Active Monitoring System” as described in our previous studies [13,14].

Nasopharyngeal swabs collected at admission were tested for influenza virus nucleic acid with an influenza universal kit via qRT-PCR (quantitative real-time PCR). Ct values were calculated to evaluate the viral load and the values with less than 35 were defined as positive. EDTA (Ethylene Diamine Tetraacetic Acid) anticoagulant whole blood and procoagulant serum were obtained simultaneously for laboratory analyses.

The study groups include the primary cohort (PH3 group), with PCR-confirmed H3N2-positive pregnant women (other subtypes excluded). Severe influenza was screened from the hospitalized patients with one of the following criteria: oxygen therapy, ICU admission, or mechanical ventilation, and divided into the PSI group (pregnant severe influenza).

Control groups consisted of non-pregnant and pregnant healthy women aged 20–40 years recruited from the population attending physical examinations in the same level of sentinel hospitals. Through questionnaire surveys and analysis, individuals with a similar socioeconomic status to the study groups, no previous exposure to influenza or respiratory symptoms (one month before examination), and no history of vaccination were included in this study. Non-pregnant healthy women served as the CN (Control Negative) group. Pregnant healthy women without infection were stratified into trimester 1st, 2nd, and 3rd and served as the pregnant negative groups (PN-T1, PN-T2, and PN-T3).

Pregnant women with H3N2 influenza virus infection were similarly categorized as PH3-T1, PH3-T2, and PH3-T3, as shown in Figure 1. The 1st, 2nd, and 3rd trimesters were defined as 1–13 weeks, 14–27 weeks, and 28–40 weeks of gestation, respectively. All participants provided written informed consent prior to enrollment.

### 2.2. Analysis of Immune Cell Proportions and Function in Peripheral Blood

Peripheral blood mononuclear cells (PBMCs) were isolated from EDTA-anticoagulated blood samples to evaluate alterations in T cell, B cell, and NK cell populations among healthy controls and pregnant women infected with influenza A H3N2 virus. All blood samples were transported to the laboratory under controlled conditions and processed within 24 h of collection. For PBMC isolation, 2–3 mL of whole blood was diluted with an equal volume of DPBS (Dulbecco’s Phosphate Buffered Saline) (Gibco, Carlsbad, CA, USA) layered onto a Ficoll-Paque premium density gradient medium (GE Healthcare, Chicago, IL, USA), and centrifuged at 600× *g* for 30 min at room temperature. After centrifugation, the white blood cell layer PBMC was carefully aspirated from the interface and washed. For flow cytometry analysis, the cells were stained with a fluorescent panel of labeled TBNK CD antibodies (CD3, CD4, CD8, CD16, CD56, CD19, CD27, CD38) (Biolegend, San Diego, CA, USA) for 30 min, followed by analysis by BD FACS Canto II flow cytometry. DAPI (4′,6-diamidino-2-phenylindole) was stained for 5 min to exclude dead cells. The proportion and activation status of different immune cell populations were analyzed, including T lymphocytes (CD3^+^), helper T cells (CD3^+^CD4^+^CD8^−^), cytotoxic T cells (CD3^+^CD4^−^CD8^+^), B lymphocytes (CD19^+^), memory B cells (CD19^+^CD27^+^CD38^+/−^), plasma cells (CD19^−^CD27^+^CD38^+^), NK cells (CD3^−^CD16^+^56^+^), and NKT cells (CD3^+^CD16^+^CD56^+^). This immunophenotyping approach enabled the comparative analysis of lymphocyte populations between healthy controls and pregnant women with or without H3N2 influenza infection.

### 2.3. Quantification of Serum Inflammatory Cytokines by CBA Technology

Procoagulated blood for serum separation was transported to the laboratory at low temperature and processed within 24 h. After separation, serum samples were aliquoted and stored at −80 °C until analysis using the CBA kit (Cytometric Bead Array, Human Th1/Th2/Th17 CBA Kit) (BD Bioscience, San Jose, CA, USA). The CBA assay quantified seven key cytokines representing distinct immune responses, with Th1-associated (IL-2, TNF-α, and IFN-γ), Th2-associated (IL-4, IL-6, and IL-10), and Th17-associated (IL-17A) immune responses. Standard curves were generated using serial dilutions of recombinant cytokine standards, and PE (phycoerythrin)-conjugated detection antibodies provided fluorescence signals proportional to cytokine concentrations. Serum samples from different groups were analyzed using flow cytometry according to the manufacturer’s instructions, and cytokine concentrations were calculated based on the standard curve. The reagents used in this study are listed in Appendix A.

### 2.4. Serum Metabolomics Profiling by UPLC-MS

Serum metabolomics from the above different groups were detected through Ultra-High-Performance Liquid Chromatography coupled with High-Resolution Mass Spectrometry (UPLC-MS). For the sample preparation, 50 µL serum was used for each sample and mixed with 200 µL ice-cold methanol containing amino acid internal standards. After vortexing vigorously for 3 min, the samples were centrifuged at 15,000 rpm for 10 min at 4 °C. Then the supernatant was transferred to 96-well plates for nitrogen drying and reconstituted in 200 µL 10% methanol for analysis. QC samples were prepared by pooling equal volumes from all serum samples.

For instrumental analysis, metabolite profiling was performed using a Thermo Scientific (Waltham, MA, USA) Vanquish Horizon UHPLC system coupled to an Orbitrap Exploris 240 mass spectrometer. The chromatographic conditions were as follows: columns accucore C18 and HILIC. Mobile phase, A: 0.1% formic acid in water and B: 0.1% formic acid in acetonitrile. Gradient: 0–1 min, 1% B; 1–3 min, 1–15% B; 3–7 min, 15–50% B; 7–110 min, 50–100% B; 10–12 min, 100% B; 12–12.5 min, 100–1% B; 12.5–15 min, 1% B. Flow rate: 0.4 mL/min; column temperature: 40 °C; injection volume: 5 µL. For mass spectrometry parameters, ionization mode: electrospray ionization (ESI) in positive and negative polarity modes. Source settings: spray voltage, ±3.5 kV. Ion transfer tube temp.: 325 °C. Vaporizer temp.: 350 °C. Sheath gas: 50 arb; aux gas: 10 arb. Full scan acquisition: resolution, 120,000 (at *m*/*z* 200). Scan range: *m*/*z* 70–1050. Automatic gain control (AGC): 3e6. Maximum injection time: 100 ms. Data-dependent MS/MS: top 10 most intense ions fragmented using higher-energy collisional dissociation (HCD) with stepped normalized collision energy (20, 30, and 40%).

For data processing, the raw data were processed using compound discoverer 3.3 (Thermo Fisher, Waltham, MA, USA) for peak alignment, retention time correction, and metabolite annotation. Databases including mzCloud, HMDB, and METLIN were used for spectral matching (mass error < 5 ppm). For quality control, the pooled QC samples were injected every 10 injections to monitor system stability. Features with relative standard deviation (RSD) >30% in QCs were excluded.

For metabolomics statistical analysis, multivariate analysis (PCA and PLS-DA) and univariate tests (Student’s *t*-test and fold change analysis) were performed using MetaboAnalyst 6.0. False discovery rate (FDR) correction (Benjamini–Hochberg) was applied to adjust *p* values. Two filtration criteria (fold change > 2 and *p* value < 0.05) were used to obtain significantly differential metabolites. A hierarchical cluster was used to cluster the differential metabolites at the sample level. At the same time, enrichment analysis of the KEGG pathway was performed on the differential metabolites. Biomarker potential was evaluated by classical ROC (receiver operating characteristic) curve and AUC (area under ROC curve) analysis. Biomarkers were screened with 0.7 < AUC < 1. An AUC greater than 0.7 was used as a screening criterion, 0.8–0.9 can be used as a potential marker, and 0.9–1 is regarded as an ideal biomarker [15].

### 2.5. Statistical Analysis

Statistical analyses for other data were conducted using SPSS V22.0 and GraphPad Prism 8.0. The CN group served as a reference group for comparisons with other experimental groups. The PH3 group was additionally compared to the PN group, and the PSI group was additionally compared to the PH3 group and PN group. For quantitative data, normality and lognormality tests were first performed. Data conforming to a normal or Gaussian distribution were analyzed using a parametric unpaired *t*-test. Nonparametric data, not conforming to a Gaussian distribution, were analyzed using a nonparametric rank sum test. Data following a Gaussian distribution were presented as the mean ± SEM, while non-Gaussian data were presented as the median with a 95% CI. A *p* value < 0.05 was considered statistically significant.

## 3. Results

### 3.1. Description of the Enrolled Population

A prospective cohort was conducted during the 2023–2024 influenza season (1 October to 31 March) in Suzhou, China, enrolling 1372 pregnant women presenting with acute respiratory infection at sentinel hospitals. Of these, 276 cases were laboratory-confirmed influenza A/H3N2 positive. A total of 241 H3N2-positive participants provided anticoagulated and serum samples. These samples were subsequently processed for various analyses. Eighty-two fresh blood samples were separated for PBMC isolation and subsequent analysis using flow cytometry. The remaining blood samples were allocated for RNA extraction and other experimental procedures. Among the 82 samples, 12 were from the first trimester, 11 were from the second trimester, and 39 were from the third trimester, labeled PH3-T1, PH3-T2, and PH3-T3, respectively. For severe influenza, a total of 20 cases were screened, and they were all in the third trimester of pregnancy.

For the control groups, 37 samples were used for analysis for CN. For the PN group, 45 were from the first trimester, 49 were from the second trimester, and 39 were from the third trimester, labeled PN-T1, PN-T2, and PN-T3, respectively. The control groups had a similar socioeconomic status to influenza-positive study groups, had no respiratory symptoms within one month before examination, and had no history of influenza vaccination due to the low vaccination rate among pregnant women. The demographic and clinical data of the participants are shown in Appendix A.

### 3.2. TBNK Immune Cell Alteration in Pregnant Women Infected with Influenza A H3N2 Virus

The results showed significant changes in immune cell populations, particularly in the second and third trimesters of pregnancy. For T cell alteration (Figure 2), a significant reduction in total CD3^+^ T cells was observed in the second (PH3-T2) and third (PH3-T3) trimesters of pregnancy influenza groups and the PSI group compared to both the control (CN) and healthy pregnancy (PN) groups. A similar decline was noted in CD4^+^ T cells in all PN groups, PH3-T2/T3, and PSI groups. Meanwhile CD8^+^ T cells were elevated in all PN groups and the first trimester of PH3 (PH3-T1) but were not observed in subsequent trimesters. The decrease in CD4^+^ T cells or increase in CD8^+^ T cells resulted in a decline in the CD4/CD8 ratio across all pregnant groups, including both healthy women and those infected with H3N2. This ratio fell below the normal value of 1.4, suggesting potential immune tolerance and compromised immune function in pregnant women, particularly those infected with H3N2.

For B cells (Figure 3), a significant decrease in total CD19^+^ B cells was observed in the PH3-T2/PH3-T3 and PSI groups compared to both the CN and PN groups. Memory B cells (CD19^+^CD27^+^CD38^+/−^) in all PH3 and PSI groups declined compared to both the CN and PN groups. However, no significant differences were noted in plasma cell populations in all groups. The loss or deficiency of memory B cells, especially in the second and third trimester of pregnancy, could lead to a diminished humoral immune response and increased susceptibility to influenza virus in pregnant women.

For NK cells (Figure 4), a significant increase in NK cells (CD3^−^CD16^+^56^+^) in the PSI group was observed compared to the PN and PH3 groups. No significant changes were observed in NKT (CD3^+^CD16^+^56^+^) cells across the different groups mostly due to interindividual variability. As an important part of the natural immune system, NK cells are the first line of defense against viral infection, especially influenza virus infection. The rise in NK cells in severe cases of influenza in pregnant women is essentially a manifestation that the body’s natural immune system is strongly activated to fight against the virus under strong viral stimulation. However, in the severe state, this activation often loses control and evolves into hyperactivation and dysfunction.

### 3.3. Cytokine Screening of Early Biomarkers for Severe Influenza Cases

We evaluated serum concentrations of seven cytokines (IL-2, IL-4, IL-6, IL-10, TNFα, IFNγ, and IL-17a) as potential biomarkers for influenza severity in pregnant women (Figure 5). The cytokine analysis was performed without trimester stratification to assess the overall disease-associated patterns.

No significant differences were observed between the non-pregnant control (CN) and pregnant negative (PN) groups (all *p* > 0.05), suggesting that pregnancy status alone does not significantly influence the baseline cytokine levels.

In pregnant women with H3N2 infection (PH3), a significant elevation in IL-6 and IL-10 levels was observed compared to both the PN and CN groups, and IFNγ was increased compared to the PN group (*p* < 0.05). Conversely, IL-17a levels were significantly decreased in the PH3 group, which suggests the co-existence of pro-inflammatory (IL-6 and IFNγ) and anti-inflammatory responses (IL-10).

The group of pregnant women with severe influenza (PSI) exhibited significantly increased levels of IL-4 and IL-6 compared to the PH3 group and an elevated IL-10 level compared to the CN group. TNFα and IFNγ levels in the PSI group also showed an upward trend but did not reach statistical significance. The results suggest that elevated levels of IL-4, IL-6, and IL-10, which are associated with Th2 cell activation, might be related to severe influenza, but the increased TNFα and IFNγ levels also need to be paid attention.

### 3.4. Serum Metabolomics Revealed Kyn/Trp as Indicated Biomarkers for Severe Influenza Cases in Pregnant Women

Serum metabolomic alterations were analyzed using UPLC-MS. Through analysis, a total of 3821 metabolites were identified in all groups. Compared to healthy non-pregnant controls, 351 differential metabolites were identified in pregnant women not infected with influenza, with 276 up-regulated and 75 down-regulated. Notably, 884 (270 up and 614 down) and 512 (265 up and 247 down) differential metabolites were detected in pregnant women with influenza and severe influenza, respectively, compared to the normal pregnancy group. A total of 352 (265 up and 87 down) differential metabolites were identified in the PSI group compared to the PH3 group (Table 1).

Figure 6A shows the PCA plot of the different groups. CN was clustered on the left, indicating metabolic homogeneity. PN overlapped partially with CN, suggesting pregnancy alone causes minor metabolic shifts. PH3 shifted rightward, clearly separated from the healthy groups, indicating infection-induced metabolic disruption. PSI partially overlapped with PH3, but downward, highlighting unique metabolic dysregulation in severe cases. Similar to the PCA plot, the heatmap of different metabolites clustered in the four groups is shown in Appendix A, in which the CN and PN groups had similar metabolites, but influenza virus infection completely altered the expression of the metabolites, resulting in significantly different patterns between the PH3/PSI group and control groups, indicating that influenza virus infection had largely altered the metabolic profile of pregnant women.

Comparative and enrichment analyses through KEGG pathway enrichment analysis revealed that the differential metabolites were predominantly enriched in amino acid metabolism (such as tryptophan metabolism and histidine metabolism), lipid metabolism (such as glycerophospholipid metabolism), and hormonal signaling pathways (such as steroid hormone biosynthesis) (Figure 6B) in the PSI group compared to the PN group. Among the pathways, the tryptophan–kynurenine (Trp–Kyn) pathway exhibited marked dysregulation in pregnant women with influenza virus infection. This was characterized by the depletion of tryptophan and concomitant accumulation of its metabolite 3-hydroxykynurenine (Figure 6C). The AUC of Trp and 3-H-Kyn reached 0.759 and 0.927, respectively, with a 95% CI of 0.65–0.856 and 0.864–0.973. Furthermore, the ratio of Kyn and Trp increased significantly, especially in severe cases (fold change >15 compared to the PN group and fold change >3 compared to the PH3 group), suggesting that it has a potential role as a biomarker for severe influenza. Most of the metabolites in the PSI group and PH3 group showed a similar pattern; however, a small number of differential metabolites were also found through comparison, as shown in Table 2. The most obvious change between the PSI and PH3 group was N,N-dimethylsphingosine, which changed more than 40 times, and other metabolites such as oleamide and itaconic acid were also identified. But the role of these metabolites in severe influenza needs further study.

## 4. Discussion

The human immune system orchestrates a sophisticated defense network against pathogenic invasions [16] while pregnancy presents a unique immunological paradox that necessitates simultaneous fetal tolerance and maternal antimicrobial defense [17,18]. This delicate equilibrium involves dynamic immune adaptations rather than generalized suppression, as evidenced by emerging insights into pregnancy-associated immunological reprogramming [19,20,21]. Our study systematically characterizes trimester-specific alterations in TBNK lymphocyte subsets and cytokine profiles during H3N2 influenza infection while revealing profound perturbations in the tryptophan–kynurenine (Trp–Kyn) pathway associated with disease severity.

Our findings demonstrated a significant immunological alteration characterized by CD4^+^ T cell decline coupled with CD8^+^ T cell expansion during mid-to-late gestation, leading to a substantially reduced CD4/CD8 ratio. This immunological shift suggests an impaired cell-mediated immunity that could potentially enhance maternal susceptibility to influenza viral infection [22]. This aligns with previous reports regarding pregnancy-associated immunological adaptations, including suppressed Th1 responses and enhanced Th2/Treg cell activity [23]. Notably, influenza infection amplified these immunological shifts, particularly in CD27^+^ memory B cell depletion during advanced pregnancy stages. Given the well-established crucial role of memory B cells in mediating rapid, high-affinity secondary antibody responses [24], this specific deficiency may partially explain the two clinically relevant phenomena in pregnant populations: an extended duration of viral shedding and a higher incidence of influenza-related complication.

Cytokine analysis revealed a distinct immunological pattern in pregnant women infected with influenza, characterized by the simultaneous elevation of both pro-inflammatory (IL-6, IFN-γ) and anti-inflammatory (IL-10) mediators; this finding challenges the conventional Th1/Th2 paradigm [25,26]. In severe influenza cases, the observed Th2-polarized cytokine profile (marked by elevated IL-4, IL-6, and IL-10 levels) is consistent with established pregnancy-associated immune modulation [27]. However, the concurrent IFN-γ elevation indicates that viral infection may override the typical gestational Th1 suppression, which was also related to the persistent elevation of NK cells in severe cases, the main source cells of IFN-γ production [28]. This immunological ambivalence may create a permissive environment for both viral persistence and cytokine-mediated tissue damage, potentially explaining the dual risk of severe influenza infection and associated obstetric complications.

The observed metabolic dysregulation within the tryptophan–kynurenine (Trp–Kyn) pathway via serum metabolomics offers novel insights into this immunological conundrum. The progressive elevation of the Kyn/Trp ratio from healthy pregnancy to severe influenza was correlated with 3-hydroxykynurenine (3-HK) accumulation, mirroring patterns seen in SARS-CoV-2-induced immune exhaustion [29,30]. Given that indoleamine 2,3-dioxygenase (IDO)-mediated tryptophan catabolism promotes both immune tolerance and oxidative stress generation [31,32], its dysregulation during pregnancy may establish a metabolic vicious cycle. Specifically, viral-induced IDO activation could simultaneously impair antiviral T-cell responses via tryptophan depletion and promote reactive oxygen species (ROS)-mediated tissue damage through 3-HK accumulation. As a 15-fold change in severe influenza compared to normal pregnancy, the ratio of Kyn and Trp has potential as a biomarker of severe influenza in pregnant women. However, there also existed some limitations due to our small sample size. In a small-sample serum metabolomics study, when using ROC curves and AUC values to evaluate biomarker discrimination performance, the statistical challenges inherent to the sample size necessitate careful consideration. Potential AUC bias requires correction by methods like bootstrapping or support from established biological pathways (e.g., KEGG enrichment analysis). It is generally considered that the value of the AUC is statistically significant when it is greater than 0.7. Here, Trp and Kyn yielded an AUC of 0.759 and 0.927, respectively, with robust confidence intervals and KEGG pathway enrichment. While demonstrating significant potential as biomarkers for severe influenza infection, validation in larger, multicenter cohorts remains essential.

Furthermore, we identified some intriguing metabolites in severe influenza cases such as the elevation of N,N-dimethylsphingosine (DMS) and itaconate. The elevation of them suggests cross-talk between sphingolipid metabolism and macrophage activation pathways. DMS, a sphingosine kinase inhibitor, exhibits notable immunomodulatory and organ-protective effects in cardiovascular and infectious diseases. In experimental chronic Chagas cardiomyopathy [33], DMS attenuated myocardial inflammation and fibrosis by inhibiting the sphingosine-1-phosphate (S1P) pathway, reducing pro-inflammatory cytokines (TNF-α, IFN-γ), and improving cardiac function. Similarly, in myocardial ischemia-reperfusion injury [34], DMS recruited regulatory T cells (Tregs) to the injured heart via the PI3K/Akt pathway, mitigating neutrophil infiltration and oxidative stress, thereby reducing the infarct size. Itaconate, a key macrophage immunometabolite, plays a pivotal role in respiratory viral infections. An et al. [35] demonstrated that quercetin induces itaconate-mediated metabolic reprogramming in alveolar macrophages, balancing M1/M2 polarization: itaconate activates the Nrf2 antioxidant pathway and inhibits NLRP3 inflammasome, shifting macrophages from pro-inflammatory M1 to reparative M2 phenotypes. This significantly reduced lung injury and viral load in respiratory syncytial virus (RSV) infection. Thus, the elevation of DMS and itaconate in severe influenza cases may represent a beneficial protective response.

The limitations of our study are as follows: One single city cohort, a small sample size in some groups such as PSI, and a cross-sectional study. Although our study identifies some promising candidates like CD4/CD8, IL-6, and Kyn/Trp, longitudinal validation in independent larger cohorts is essential to confirm their clinical significance. Mechanistic studies are also warranted to determine whether Trp–Kyn dysregulation directly contributes to pathogenesis or merely reflects compensatory anti-inflammatory responses in pregnancy. Future research utilizing a larger, multicenter cohort is recommended to further elucidate the implications of these findings for influenza immune response efficacy and potential biomarkers used for the severity of influenza infection for the purpose of developing strategies to prevent severe disease in pregnant women.

## 5. Conclusions

The observed decline in the CD4/CD8 ratio and reduction in memory B cells in pregnant women with influenza infection suggest weakened cellular and humoral immune defenses, potentially rendering pregnant women more vulnerable to influenza, particularly during the middle and late stages of pregnancy. The elevation of both pro-inflammatory and anti-inflammatory cytokines indicates a complex interplay of excessive inflammation and immunosuppression in pregnant women with influenza infection. The increase in Th2 cytokines in severe influenza cases points towards a potential link between a Th2 cell-mediated immune response and disease severity in this population, as well as the elevation of NK cells. Serum metabolomics revealed the ratio of Kyn/Trp as indicated potential biomarkers for severe influenza cases in pregnant women. However, further research in larger and multicenter cohorts is needed to validate the reliability, and the precise mechanisms underlying these immune alterations and their implications for influenza susceptibility and disease severity in pregnant women also remain elucidated.

## Figures and Tables

**Figure 1 diseases-13-00182-f001:**
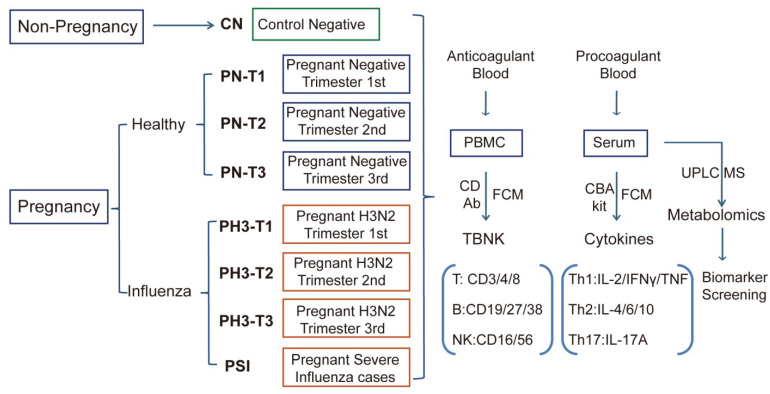
Scheme and implementation of this study. Non-pregnant healthy (CN), pregnant healthy (PN)/influenza women (PH3), and severe influenza cases of pregnant women (PSI) were included in this study, and corresponding biological samples were collected. Anticoagulants were used to detect the proportion and function of TBNK immune cells, and procoagulant serum was used to detect cytokines and serum metabolomics.

**Figure 2 diseases-13-00182-f002:**
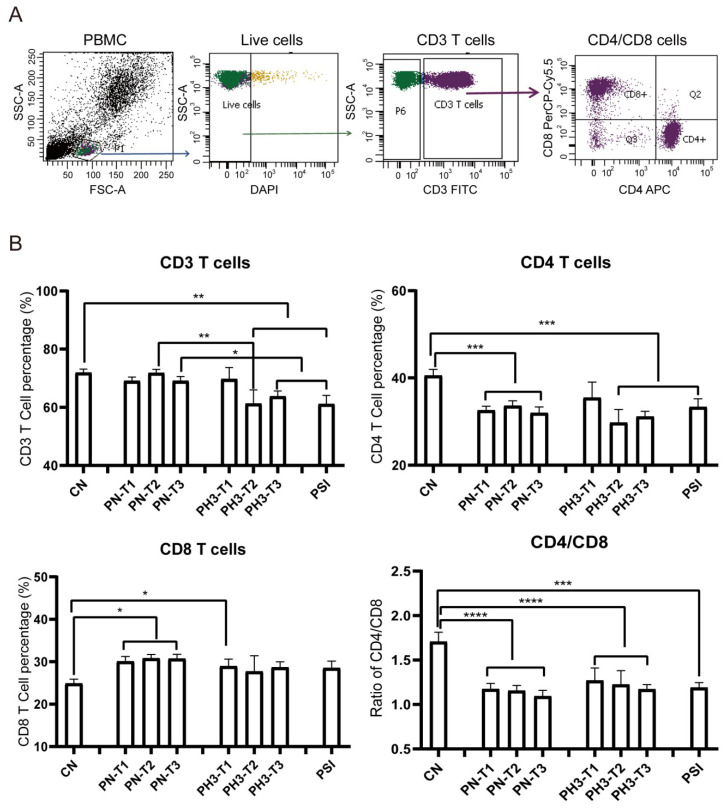
T cell alteration in pregnant women infected with influenza A virus. (**A**) Flow chart for T cell analysis. CD3 cells were circled on the basis of live cells, and CD4/CD8 cells were further conducted on CD3 cells. (**B**) Statistical analysis of T cells, including CD3^+^ T cells, CD4^+^ T cells, and CD8^+^ T cells, and the ratio of CD4/CD8 among the 8 groups. *: *p* < 0.05; **: *p* < 0.01; ***: *p* < 0.001; ****: *p* < 0.0001.

**Figure 3 diseases-13-00182-f003:**
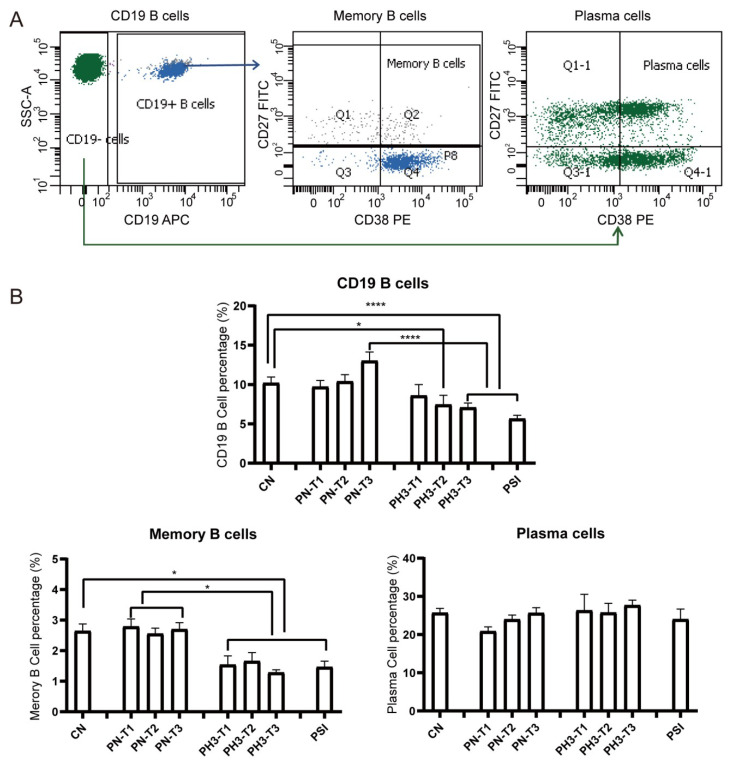
B cell alteration in pregnant women infected with influenza A virus. (**A**) Flow chart for B cell analysis. Memory B cells were based on CD19^+^ cells and CD27^+^ CD38^+/−^. Plasma cells were based on CD19^−^ cells and CD27^+^ CD38^+^. (**B**) Statistical analysis of B cells, including CD19^+^ B cells, CD19^+^CD27^+^CD38^+/−^ memory B cells, and CD19^−^CD27^+^CD38^+^ plasma cells among the 8 groups. *: *p* < 0.05; ****: *p* < 0.0001.

**Figure 4 diseases-13-00182-f004:**
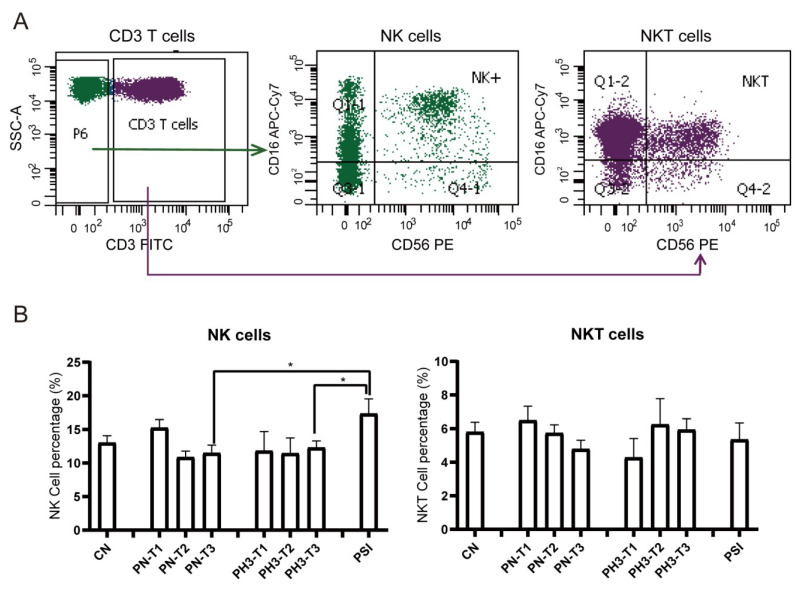
NK cell alteration in pregnant women infected with influenza A virus. (**A**) Flow chart for NK cell analysis. NK cells were based on CD3^−^ cells and CD16^+^/CD56^+^. NKT cells were based on CD3^+^ cells and CD16^+^/CD56^+^. (**B**) Statistical analysis of NK cells, including CD3^−^CD16^+^CD56^+^ NK cells and CD3^+^CD16^+^CD56^+^ NKT cells among the 8 groups. *: *p* < 0.05.

**Figure 5 diseases-13-00182-f005:**
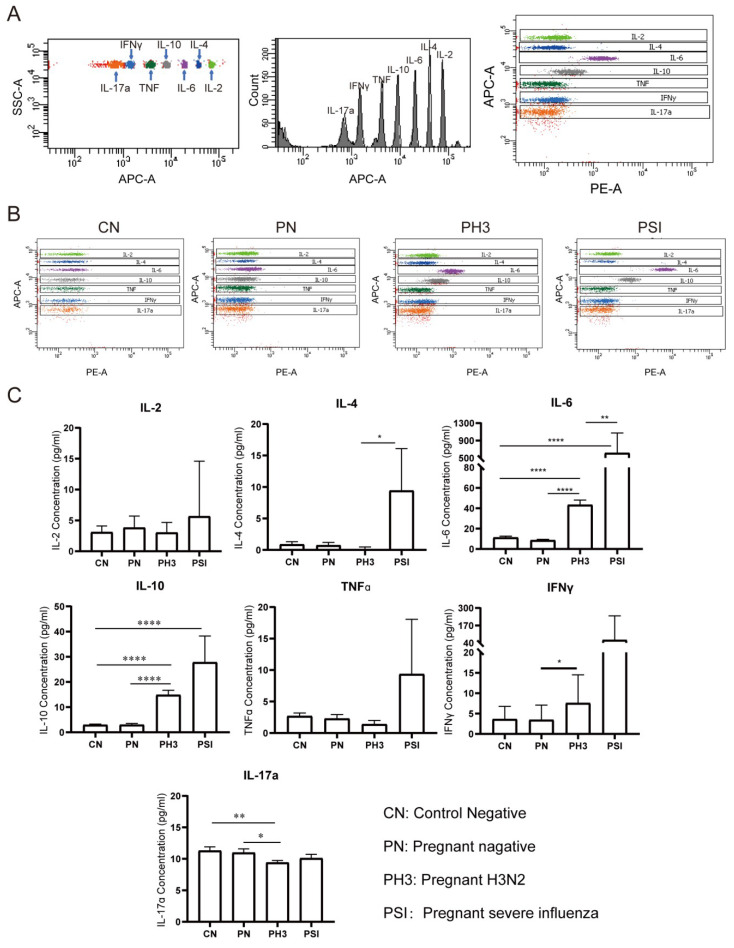
Cytokine screening of early biomarkers for severe cases. (**A**) The principle of detecting different cytokines by the adaptor of APC and detector of PE. (**B**) The representative flow chart of the cytokines in each group. (**C**) Statistical analysis of the cytokines, including IL-2, IL-4, IL-6, IL-10, TNFα, IFNγ, and IL-17a. *: *p* < 0.05; **: *p* < 0.01; ****: *p* < 0.0001.

**Figure 6 diseases-13-00182-f006:**
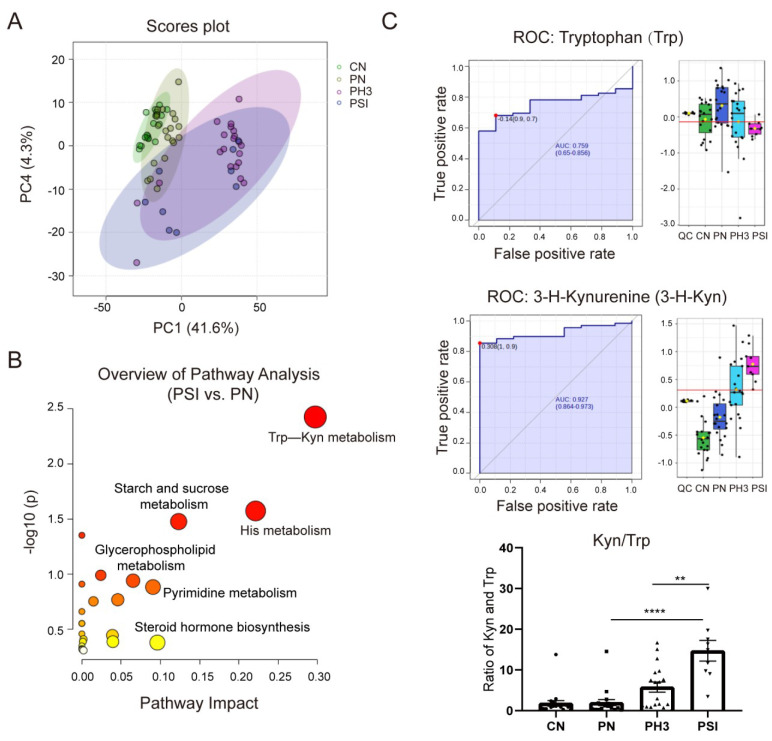
Serum metabolomics revealed Trp–Kyn as indicated potential biomarkers for severe influenza cases in pregnant women. (**A**) Principal component analysis (PCA) plot of filtered metabolomics data of different groups. (**B**) Bubble plot of statistically significant metabolic pathways enriched in PSI group vs. PN group. (**C**) Area under ROC Curve (AUC) of tryptophan (Trp) and 3-hydroxykynurenine (3-H-Kyn) and ratio of Kyn and Trp. **: *p* < 0.01; ****: *p* < 0.0001.

**Table 1 diseases-13-00182-t001:** Number of different metabolites identified in serum metabolomics.

Groups	Up-Regulated	Down-Regulated	Non-Significant
PN vs. CN	276	75	3470
PH3 vs. PN	270	614	2937
PSI vs. PN	265	247	3309
PSI vs. PH3	265	87	3469

Legends: This table shows the numbers of different metabolites including up/down-regulated and non-significantly changed metabolites in the four groups. CN: Control Negative; PN: pregnant negative; PH3: pregnant H3N2; PSI: pregnant severe influenza.

**Table 2 diseases-13-00182-t002:** Enriched metabolites with KEGG pathway in PSI group vs. PH3 group.

Name	HMDB	PubChem	KEGG	FC (PSI vs. PH3)	log2(FC)	*p* Value	FDR
N,N-Dimethylsphingosine	HMDB0013645	5282309	C13914	40.605	5.347	1.47 × 10^−15^	1.87 × 10^−12^
Oleamide	HMDB0002117	5283387	C19670	12.351	3.627	8.93 × 10^−9^	4.21 × 10^−6^
Adipic acid	HMDB0000448	196	C06104	3.887	1.959	1.73 × 10^−7^	3.31 × 10^−5^
Amide C18	HMDB0034146	31292	C13846	3.846	1.943	1.23 × 10^−8^	4.35 × 10^−6^
Testosterone sulfate	HMDB0002833	119207	NA	3.697	1.886	1.56 × 10^−7^	3.31 × 10^−5^
Styrene	HMDB0034240	7501	C07083	3.644	1.866	1.34 × 10^−7^	3.19 × 10^−5^
Phaseic acid	HMDB0302844	5281527	C09707	3.643	1.865	6.36 × 10^−8^	1.87 × 10^−5^
Itaconic acid	HMDB0002092	811	C00490	2.835	1.503	2.43 × 10^−4^	1.75 × 10^−2^
Linoleamide	HMDB0062656	6435901	NA	2.590	1.373	3.56 × 10^−5^	3.60 × 10^−3^

Legends: This table shows enriched metabolites with KEGG pathway in PSI group vs. PH3 group. HMDB: human metabolome database; KEGG: Kyoto Encyclopedia of Genes and Genomes; FC: fold change; FDR: false discovery rate.

## Data Availability

The datasets used and analyzed during the current study are available from the corresponding author upon reasonable request.

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
