# Peer review of "Immune Response and Serum Biomarker Screening in Pregnant Women with Influenza A Virus Infection: Insight into Susceptibility and Disease Severity"

_diseases, 2025, doi:10.3390/diseases13060182_

Round 1
Reviewer 1 Report
Comments and Suggestions for Authors
In the present paper, the authors report the “Immune Response and Serum Biomarker Screening in 2 Pregnant Women with Influenza A Virus Infection: Insight 3 into Susceptibility and disease Severity”. I would recommend the acceptance of this manuscript after minor revisions. Here are the following comments.
- What actions were taken to ensure that control groups, including pregnant and non-pregnant healthy women, were adequately matched to the infected cohort regarding confounding variables such as socioeconomic status, comorbidities, prior influenza exposure, and vaccination history?
- Since peripheral blood mononuclear cells (PBMCs) were not collected from the severe influenza (PSI) group, how can you support any claims regarding immune cell dysregulation in severe disease cases?
- How can you recommend the CD4/CD8 ratio, IL-6, and Kyn/Trp for clinical use without validation in larger or independent cohorts?
- How can you justify the generalizability of your findings from a cohort-based in a single city?
- What statistical justifications support the reliability of ROC analysis for the Kyn/Trp ratio and other metabolomic markers, considering that only 11 severe cases were included in the study?
- Did you quantitatively measure the influenza viral load to account for differences in immune responses?
Author Response
Comment 1: What actions were taken to ensure that control groups, including pregnant and non-pregnant healthy women, were adequately matched to the infected cohort regarding confounding variables such as socioeconomic status, comorbidities, prior influenza exposure, and vaccination history?
Response 1: We addressed this by: Recruiting controls from the same level hospitals and similar socioeconomic backgrounds. Excluding individuals with recent respiratory symptoms or vaccination history. We also added a table discribed the detailed demographic and clinical data shown in Supplementary Table 2. (Revised: Section 2.1, lines 84-87; Section 3.1, line 201-205, and Suppl. Table 2)
Comment 2: Since peripheral blood mononuclear cells (PBMCs) were not collected from the severe influenza (PSI) group, how can you support any claims regarding immune cell dysregulation in severe disease cases?
Response 2: We reanalyzed the clinical data of the influenza cases in pregnancy, and screened out 20 severe cases according to the cretiria of severe influenza, from which PBMCs had also been isolated and subjected to flow cytometry in previous experiment. So the PBMC data of severe cases (PSI group) were also analyzed. The results were shown in Figure 2-4. The immune response in PSI group was mostly similar to PH3-T3 in T cells and B cells, but NK cells were significantly increased in PSI group compared to PN and PH3 groups, which might be correlated with the hyperactivation and immune dysfunction following viral infection of this group. (Revised: Figures 2-4 and Results 3.1-3.2)
Comment 3: How can you recommend the CD4/CD8 ratio, IL-6, and Kyn/Trp for clinical use without validation in larger or independent cohorts?
Response 3: We acknowledge the limitation of our study in one single city and the small sample size in some groups. We have added the limitation in the discussion part in section 4, line 400-408 that longitudinal validation in independent larger cohorts is essential to confirm their clinical significance. (Revised: section 4, line 400-408)
Comment 4: How can you justify the generalizability of your findings from a cohort-based in a single city?
Response 4: This was similar to comment 3, and we acknowledge the limitation, and future multicenter studies are needed to validate generalizability of these candidates. We have added in the discussion part seen in section 4, line 405-407. (Revised: section 4, line 405-407)
Comment 5: What statistical justifications support the reliability of ROC analysis for the Kyn/Trp ratio and other metabolomic markers, considering that only 11 severe cases were included in the study?
Response 5: About the biomarker screening through ROC and AUC analysis, we have added the criteria in the method 2.4, line 171-174, and results 3.4, line 310-311. In the discussion part, we also discussed the biomarker evaluation performance in small sample size. It is generally considered that the value of AUC is statistically significant when it is greater than 0.7. Here, Trp and Kyn yielded AUC of 0.759 and 0.927, respectively, with robust 95% confidence intervals and KEGG pathway enrichment (Figure 6B). While demonstrating significant potential as biomarkers for severe influenza infection, validation in larger, multicenter cohorts remains essential. (Revised: method 2.4, line 171-174, results 3.4, line 310-311 and section 4, line 372-380)
Comment 6: Did you quantitatively measure the influenza viral load to account for differences in immune responses?
Response 6: Influenza viral load was assessed via qRT-PCR Ct values (Ct <35 defined as positive). (Methods 2.1, line 73-75.) However, the correlation analysis between viral load and immune response of patients was not performed. But the blood samples were collected at the time of admission, reflecting the immune response at the peak of virus infection. (Revised: Methods 2.1, line 73-75.)
Reviewer 2 Report
Comments and Suggestions for Authors
The presented manuscript by a group of co-authors is about risks to pregnant women caused by seasonal influenza, while the immunological mechanisms underlying their heightened disease susceptibility remain incompletely characterized. This study employed multiparametric immunophenotyping and metabolic profiling to investigate cellular immunity, cytokine dynamics, and serum biomarkers in H3N2-infected pregnant women across gestational stages. These researchers through integrated flow cytometric analysis of peripheral blood mononuclear cells (PBMCs), multiple cytokine quantification, and LC-MS-based serum metabolomics, they compared immunological parameters, serum cytokines, and metabolites across trimesters in normal and infected pregnant women. Metabolomic profiling identified dysregulation of the tryptophan-kynurenine (Trp-Kyn) pathway, with a 15-fold increase in the Kyn/Trp ratio in severe influenza. According to these co-authors, these results elucidate two synergistic pathophysiological axes - immune dysregulation and tryptophan metabolism alteration - that potentially drive adverse outcomes. The identified biomarker panel (CD4/CD8 ratio, IL-6, Kyn/Trp) shows clinical promise for early risk stratification in high-risk pregnancies with influenza infection. However, according to them, further research is needed to elucidate the precise mechanisms underlying these immune alterations and their implications for influenza susceptibility and disease severity in pregnant women. Generally, the manuscript is well written and full of several updated citations.
After checking the figures, I would suggest possibly the improvement of some of the graphics, which resolution seems very low and possibly separate them in different Figures. Generally, the manuscript is a valuable one, because the quality and quantity of the provided information for the scientific community is good.
Author Response
Comment 1: After checking the figures, I would suggest possibly the improvement of some of the graphics, which resolution seems very low and possibly separate them in different Figures. Generally, the manuscript is a valuable one, because the quality and quantity of the provided information for the scientific community is good.
Response 1: Thanks very much for your suggestions. We have improved the resolution of all the figures, and separated previous Figure 6 to two figures (Figure 6 and Supplementary Figure 1) to show the results more clearly.
Reviewer 3 Report
Comments and Suggestions for Authors
The primary objective of this study was to investigate the complex interactions between immune cell alterations, cytokine profiles, and serum biomarkers in pregnant women infected with the influenza virus. However, the authors need to address the following comments:
- Tables 1 and 2 do not correspond to their titles. Please correct them.
- Add table footers to all tables
- Describe the inclusion and exclusion criteria for the participants included in the study.
- Add a table with demographic and clinical data for your participants. It is important to know the characteristics of your study groups; for example, do they have a record of prior influenza vaccination? Or, are there comorbidities that could interfere with the results obtained?
- N, N-Dimethylsphingosine had the most significant p-value, this is found within the Trp-Kyn pathway; please expand on this in the discussion
- Please add a paragraph with the limitations of your study.
Author Response
Comment 1: Tables 1 and 2 do not correspond to their titles. Please correct them.
Response 1: We have corrected the tables. Previous Table 1 was changed to Supplementary Table 1, and previous Table 2 was changed to Table 1.
Comment 2: Add table footers to all tables.
Response 2: We have added footers to all tables, including Table 1 and 2, Supplementary Table 1 and 2.
Comment 3: Describe the inclusion and exclusion criteria for the participants included in the study.
Response 3: We have described the inclusion and exclusion for the participants as seen in section 2.1, line 77- 81 for PH3 and PSI groups, line 84-87 for CN and PN groups.
Comment 4: Add a table with demographic and clinical data for your participants. It is important to know the characteristics of your study groups; for example, do they have a record of prior influenza vaccination? Or, are there comorbidities that could interfere with the results obtained?
Response 4: We have added a table with demographic and clinical data for our participants, seen in Supplementary Table 2, including sympotome, vaccination in the past year, influenza related serious outcomes, comorbidities, hospital level for consultation of all the participants.
Comment 5: N, N-Dimethylsphingosine had the most significant p-value, this is found within the Trp-Kyn pathway; please expand on this in the discussion.
Response 5: We have added a paragraph about N, N-Dimethylsphingosine (DMS) in the discussion part seen in Section 4, line 381-391.
Comment 6: Please add a paragraph with the limitations of your study.
Response 6: We have added a paragraph about the limitations of our study in the discussion part, line 400-408.
Reviewer 4 Report
Comments and Suggestions for Authors Seasonal influenza epidemics cause substantial global morbidity and mortality, contributing to a significant economic burden. This study aims to investigate the complex interactions between immune cell alterations, cytokine profiles, and serum biomarkers in pregnant women infected with the influenza virus. Through integrated flow cytometric analysis of peripheral blood mononuclear cells (PBMCs), multiple cytokine quantification, and LC-MS-based serum metabolomics, the authors compared immuno-logical parameters, serum cytokines, and metabolites across trimesters in normal and infected pregnant women. They found reduced CD4+/CD8+ T cell ratios, diminished CD27+ memory B-cell population in H3N2-infected pregnant women, and elevated Th2-skewed cytokines (IL-4, IL-6, IL-10) in severe cases. Through metabolomic profiling, they identified dysregulation of the tryptophan-kynurenine (Trp-Kyn) pathway, with a 15-fold increase in the Kyn/Trp ratio in severe influenza. These results elucidate two synergistic pathophysiological axes - immune dysregulation and tryptophan metabolism alteration - potentially driving adverse outcomes. The identified biomarker panel (CD4/CD8 ratio, IL-6, Kyn/Trp) shows clinical promise for early risk stratification in high-risk pregnancies with influenza infection. This study provided an essential novel scientific basis for developing targeted public health strategies to prevent and manage influenza infections in pregnant women in the post-pandemic era. However, before publication, the following comments should be addressed:- Line 128: Table 1 content is incorrect. It belongs to Table 2.
- Lines 260-264: Please elaborate on “351 different metabolites identified in normal pregnant women”. Does this mean that 351 metabolites were identified only in normal pregnant women, not in any infected women? It does not look like this in Figure 6A.
- Line 267: Figure 6A makes counting the number shown in each group difficult. A supplemental Table showing the details correlating to Figure 6B would be more informative.
- Line 267: Figure 6B is difficult to see. A supplemental Table showing the details correlating to Figure 6B would be more informative.
- Line 270: Table 2 content is incorrect. It belongs to Table 1.
- Line 270: In Figure 6D, “Tryptophan/Trp “means the ratio, or it means “Tryptophan (Trp).” Please clarify. There is the same question regarding “3-H-Kynurenione/Kyn”.
- Line 291: the spelling of Dimethylsphingosine was separated into two lines incorrectly.
- Line 332-334: The concomitant elevation of N,N-dimethylsphigosin (DMS), and itaconic acid is exciting. Please elaborate. It will make the findings more interesting.
Author Response
Comment 1: Line 128: Table 1 content is incorrect. It belongs to Table 2.
Response 1: We have corrected the tables. Previous Table 1 was changed to Supplementary Table 1, and previous Table 2 was changed to Table 1.
Comment 2: Lines 260-264: Please elaborate on “351 different metabolites identified in normal pregnant women”. Does this mean that 351 metabolites were identified only in normal pregnant women, not in any infected women? It does not look like this in Figure 6A.
Response 2: 351 were the different metabolites between PN and CN groups, including 276 up-regulated and 75 down-regulated, and 3470 metabolites with non-significant changed. Line 280-281 and Table 1.
Comment 3: Line 267: Figure 6A makes counting the number shown in each group difficult. A supplemental Table showing the details correlating to Figure 6B would be more informative.
Response 3: Figure 6A showed the PCA plot of the four groups. PCA (Principal Component Analysis) is an unsupervised dimensionality reduction algorithm used to simplify complex metabolomics data. It compresses metabolomic data (e.g., thousands of metabolites) into 2–3 principal components (PC1/PC2/PC3) for intuitive 2D/3D plotting. In Figure 6A, CN was clustered on the left, indicating metabolic homogeneity. PN overlapped partially with CN, suggesting pregnancy alone causes minor metabolic shifts. PH3 shifted rightward, clearly separated from healthy groups, indicating infection-induced metabolic disruption. PSI partially overlapped with PH3, but downward, highlighting unique metabolic dysregulation in severe cases. Section 3.4, line 286-291.
Comment 4: Line 267: Figure 6B is difficult to see. A supplemental Table showing the details correlating to Figure 6B would be more informative.
Response 4: Figure 6B are now separated from previous Figure 6 to Supplementary Figure 1, and its resolution was improved a lot. The 100 top significant metabolites were clustered into this heatmap, and the m/z of each metabolite was shown in the right of the figure. This figure showed the general metabolic profile of the different groups, in which significant different patterns was displayed between PH3/PSI group and control groups, indicated that influenza virus infection had largely altered the metabolic profile of pregnant women. Section 3.4, line 291-296 and Supplementary Figure 1.
Comment 5: Line 270: Table 2 content is incorrect. It belongs to Table 1.
Response 5: We have corrected the tables. Previous Table 1 was changed to Supplementary Table 1, and Previous Table 2 was changed to Table 1.
Comment 6: Line 270: In Figure 6D, “Tryptophan/Trp “means the ratio, or it means “Tryptophan (Trp).” Please clarify. There is the same question regarding “3-H-Kynurenione/Kyn”.
Response 6: Figure 6D are now changed to Figure 6C. “Tryptophan/Trp” has changed to “Tryptophan (Trp)”, “3-H-Kynurenione/Kyn” has changed to “3-H-Kynurenione (3-H-Kyn)”. Figure 6C.
Comment 7: Line 291: the spelling of Dimethylsphingosine was separated into two lines incorrectly.
Response 7: We have corrected this error now in line 323.
Comment 8: Line 332-334: The concomitant elevation of N,N-dimethylsphigosin (DMS), and itaconic acid is exciting. Please elaborate. It will make the findings more interesting.
Response 8: We have expanded the discussion about DMS and itaconic acid in the discussion part, line 387-405.
Reviewer 5 Report
Comments and Suggestions for Authors
In this manuscript, the authors sought to profile the immune cell and metabolite profiles of pregnant women infected with influenza, stratified by severity of clinical symptoms and trimester of pregnancy. PBMC profiling was done by FACS while metabolites were profiled by mass spectrometry. The methods utilized are standard in the field and the smallest groups have 11 samples, which should be a sufficient sample size for most conclusions. The authors claim that there are alterations in both B and T cell profiles and elevated cytokines in severe infections, but this is difficult to judge without seeing the data as mentioned below. For metabolic profiling, N,N-dimethyl sphingosine was upregulated more than 40x and oleamide over 12x. Several other metabolites were altered 2-3x. The significance of these changes is left as a subject of future work. Overall, this seems like a promising study, but the manuscript must be resubmitted with the data for interpretation.
Major comments:
Tables 1 and 2 are switched in the paper
Minor text corrections:
Line 67: place a space before the parentheses
Change the abbreviation for microliter and milliliter for a small or capital L for consistency, I would recommend the large capital L
Combine lines 194-195 to the paragraph below so it is not a single-sentence paragraph
Lines 250 and 307: change the double spacing between words to single spaces
Line 274: bu should be "by"
Author Response
Comment 1: Major comments: Tables 1 and 2 are switched in the paper.
Response 1: We have corrected the tables. Previous Table 1 was changed to Supplementary Table 1, and previous Table 2 was changed to Table 1.
Comment 2: Minor text corrections: Line 67: place a space before the parentheses.
Response 2: We have corrected this now in line 66.
Comment 3: Change the abbreviation for microliter and milliliter for a small or capital L for consistency, I would recommend the large capital L.
Response 3: We have change the abbreviation for microliter as µL in section 2.4.
Comment 4: Combine lines 194-195 to the paragraph below so it is not a single-sentence paragraph.
Response 4: We have corrected this. This sentence was now in Line 207-208 following the paragraph.
Comment 5: Lines 250 and 307: change the double spacing between words to single spaces.
Response 5: We have corrected this now in line 268 and 343.
Comment 6: Line 274: bu should be "by".
Resppnse 6: We have corrected this. This belonged to the Figure legend of Figure 6B, we have changed Figure 6B to Supplementary Figure 1.
Round 2
Reviewer 3 Report
Comments and Suggestions for Authors
Thanks to the authors.
They satisfactorily answered my questions.
Reviewer 4 Report
Comments and Suggestions for Authors
The authors addressed all my critiques adequately. The revised manuscript is significantly improved.
Reviewer 5 Report
Comments and Suggestions for Authors
In this resubmission, the authors have made several organisational and text changes. I believe that the edits have improved the manuscript. My previous comments have been sufficiently addressed. Overall, while a major mechanistic explanation of metabolite profiles is still lacking, the paper provides a consideration for future areas of research regarding improvement of healthcare during pregnancy.
I would just change a bit of the supplemental table legends as elaborated below.
Supplementary Table 1: change the tense from "were" to "are listed" or one can also just delete these three words and end with "...number of reagents."
Supplementary Table 2: change the tense from "showed" to "This table shows" or one can also just delete these three words and start with "The demographic and..."